

# Change of niche in guanaco (*Lama guanicoe*): the effects of climate change on habitat suitability and lineage conservatism in Chile

Andrea G. Castillo[1], Dominique Alò[1,2], Benito A. González[3] and Horacio Samaniego[1]

[1] Laboratorio de Ecoinformatica, Universidad Austral de Chile, Valdivia, Los Ríos, Chile
[2] Pontificia Universidad Católica de Chile, Departamento de Ecología, Santiago, Chile
[3] Laboratorio de Ecología de Vida Silvestre/Facultad de Ciencias Forestales y de la Conservación de la Naturaleza, Universidad de Chile, Santiago, Chile

Corresponding author
Horacio Samaniego,
horacio@ecoinformatica.cl

## ABSTRACT

**Background:** The main goal of this contribution was to define the ecological niche of the guanaco (*Lama guanicoe*), to describe potential distributional changes, and to assess the relative importance of niche conservatism and divergence processes between the two lineages described for the species (*L.g. cacsilensis* and *L.g. guanicoe*).

**Methods:** We used maximum entropy to model lineage's climate niche from 3,321 locations throughout continental Chile, and developed future niche models under climate change for two extreme greenhouse gas emission scenarios (RCP2.6 and RCP8.5). We evaluated changes of the environmental niche and future distribution of the largest mammal in the Southern Cone of South America. Evaluation of niche conservatism and divergence were based on identity and background similarity tests.

**Results:** We show that: (a) the current geographic distribution of lineages is associated with different climatic requirements that are related to the geographic areas where these lineages are located; (b) future distribution models predict a decrease in the distribution surface under both scenarios; (c) a 3% decrease of areal protection is expected if the current distribution of protected areas is maintained, and this is expected to occur at the expense of a large reduction of high quality habitats under the best scenario; (d) current and future distribution ranges of guanaco mostly adhere to phylogenetic niche divergence hypotheses between lineages.

**Discussion:** Associating environmental variables with species ecological niche seems to be an important aspect of unveiling the particularities of, both evolutionary patterns and ecological features that species face in a changing environment. We report specific descriptions of how these patterns may play out under the most extreme climate change predictions and provide a grim outlook of the future potential distribution of guanaco in Chile. From an ecological perspective, while a slightly smaller distribution area is expected, this may come with an important reduction of available quality habitats. From the evolutionary perspective, we describe the limitations of this taxon as it experiences forces imposed by climate change dynamics.

## INTRODUCTION

Human induced climate change is imposing severe challenges to the equilibrium of natural ecosystem functioning (*IPCC, 2013*). Organisms will either have to face extinctions or adapt (*Berg et al., 2010*) by altering their seasonal activities, home ranges, migratory patterns, abundances, and interspecific interactions (*Lenoir et al., 2008*; *Araújo, Thuiller & Yoccoz, 2009*; *Elith & Leathwick, 2009*; *Pecl et al., 2017*). The global rise of temperatures will likely accelerate extinction risks and threaten up to one in every six species (*Urban, 2010*). Studies on the impact of climate change on ungulates have shown that changes in distribution ranges include altitudinal shifts in mountain environments (*Mason et al., 2014*) and distributional shifts towards equivalent habitats (*Hu & Jiang, 2011*). In more extreme cases, local extinctions will be driven by environmental aridization (*Thuiller et al., 2006*; *Duncan et al., 2012*).

In Chile, current projections of greenhouse gas (GHG) emissions proposed by the Intergovernmental Panel on Climate Change (IPCC) indicate that temperature will increase in a North to South gradient (*IPCC, 2013*). A large 2.5 °C increase is expected in the Altiplano under the most extreme emission scenario and a milder 0.5 °C increase is projected in the southern region of Magallanes for the period 2031–2050. Additionally, a 10–15% decrease in precipitation is expected in the middle of the country (between 25 and 45°S), while forecasting a 5% rainfall increase in Patagonia and a similar snowfall decrease in the Magallanes region (*Rojas, 2012*).

The guanaco (*Lama guanicoe*) is both the most abundant native ungulate and the largest (120 kg) artiodactyl in South America (*Franklin, 1982*). The species is widely distributed throughout the Southern Cone, inhabiting cold, arid, and semiarid environments from sea level up to 5,000 m a.s.l. extending from northwestern Peru to Tierra del Fuego and Isla Navarino in the southern tip of the continent, with small populations roaming east of the Andes in the arid Chaco of Bolivia and Paraguay (*Franklin, 1982*; *González et al., 2006*). The highest population densities are found in the Andes and in Patagonia (*Baldi et al., 2016*). The species is characterized by specific anatomical, physiological, and reproductive adaptations to thrive and survive in arid environments despite the intense competition with livestock and severe degradation of their habitat (*González et al., 2013*; *Marin et al., 2013*; *Baldi et al., 2016*). The guanaco has a defined ecological role in each of its ecoregions either controling vegetation growth or dispersing seeds (*González et al., 2006*). These characteristics make the guanaco an important element within the tropic chain. For instance, it is the main prey of the puma (*Puma concolor*) (*Franklin et al., 1999*) and is the target of scavengers such as the chilla fox (*Lycalopex griseus*), the culpeo fox (*Lycalopex culpaeus*), and the Andean condor (*Vultur gryphus*) among others (*Travaini et al., 2001*; *González et al., 2006*).

Two distinct subspecies of guanaco (*L.g. cacsilensis* and *L.g. guanicoe*) have been proposed based on genetic studies. *L.g. cacsilensis* is distributed to the west of the Central Andean Plateau throughout Peru and the northern tip of Chile with occurrences mostly explained by elevation and precipitation seasonality. *L.g. guanicoe* is found on the southeastern slope of the Andes, ranging throughout Patagonia and Tierra del Fuego with occurrences mostly explained by annual precipitation, precipitation seasonality and grass cover (*González et al., 2013*; *Marin et al., 2013, 2017*). The geographical limit between the northwestern and southeastern lineages has been proposed to occur around 31°S in Chile (*Marin et al., 2017*) and the significant genetic structure found among the two guilds has led to recommend to classify the two lineages as evolutionary significant units (ESUs) following *Moritz's (1994)* criteria (*González et al., 2013*; *Marin et al., 2013*). However, the two lineages are not completely separated from each other. In fact, some populations have individuals of both lineages forming zones of mixed genetic heritage (*Marin et al., 2013*). The distribution of this mixed population is predicted to occur at the south end of the Altiplano, between 26° and 32°S approximately, and it is better explained by annual precipitation and precipitation seasonality (*González et al., 2013*). As reported by *Marin et al. (2013)*, the Andean plateau could have acted as a biogeographical and ecological barrier fostering vicariance processes that may be at the origin of the current distribution of guanaco lineages. It is presumed that climate changes that occurred in the past allowed the establishment of populations over this geographic barrier, with periods of connectivity and isolation allowing the establishment of populations with mixed genetic heritage (*Marin et al., 2013*).

While the discontinuity of the current geographical distribution of guanaco is mostly a consequence of recent human activities (*González et al., 2006*), the macroevolutionary processes leading to lineage divergence in guanaco should be taken into consideration when deciding on the conservation actions required, as it has been discussed elsewhere for other species (*Hu et al., 2015*). Current threats are mostly related to high competition for fodder with cattle and introduced mammals (*Mason et al., 2014*); predation by feral dogs, illegal hunting, and the reduction of available habitat due to the intensification of agriculture (*González et al., 2006*; *Baldi et al., 2016*). The 14.5 million hectares protected by the Chilean system of protected areas (PA) does not cover the entire species range (*Baldi et al., 2016*), prompting important questions regarding the future distribution of guanaco. Mostly, *L. guanicoe* is well adapted to a wide variety of habitats (*González et al., 2006, 2013*). However, at the intraspecific level, each lineage may respond differently to changes. As evidence from guanaco's natural history indicates, past changes in climate have clearly influenced the geographic distribution of this species, particularly in the Altiplano and Puna where guanacos and vicuñas (*Vicugna vicugna*) compete for resources since the Holocene (*Marin et al., 2013, 2017*). Thus, under the possible climate change scenarios in Chile, we expect *L.g. cacsilensis* to expand (or to shift) southward and overlap with the mixed genetic heritage population. On the other hand, we predict that *L.g. guanicoe*, characterized by a wider climatic tolerance (*González et al., 2013*), should mostly maintain its current geographic distribution.

From an evolutionary perspective, given the existence of these two lineages and the repeated suggestions of their consideration as ESUs (*Marin et al., 2013*; *Baldi et al., 2016*), it would be of great interest to evaluate the state of conservation of their niche to support with new evidence this classification. From such perspective, it becomes relevant to assess whether phylogenetic niche conservatism (PNC), the tendency of closely related species to differ less ecologically than expected by chance, or otherwise, phylogenetic niche divergence (PND), the tendency of closely related species to differ more ecologically than expected by chance may prevail under current and predicted niche segregation patterns under future climate change (*Pyron et al., 2015*; *Meynard et al., 2017*).

Based on the latest projections of climate change in the region (*Rojas, 2012*; *IPCC, 2013*) and the understanding of *L. guanicoe* taxonomy and life history, we developed models based on niche theory to assess the impact of climate change on guanaco's ESUs. By modeling the niche of *L. guanicoe* and its lineages we here: (a) estimated their current geographic distribution based on bioclimatic variables; (b) predicted their future distribution based on the projections of the best and worst climate change scenario at two different time frames (2050, 2070); (c) quantified the area predicted to be gained, lost, or remain stable in the future for both guanacos lineages and mixed population; (d) evaluated and compared how much of the Chilean PA will overlap with the future distribution area calculated for guanacos; and (e) explored the existence of niche conservatism or divergence between *L. guanicoe* lineages, in terms of their niche equivalence (*Graham et al., 2004*) and similarity (*Peterson, Soberón & Sánchez-Cordero, 1999*).

## METHODS

### Species occurrence data

We built a guanaco occurrence dataset of 3,321 records by complementing previous work by the authors with 359 additional records (*González et al., 2013*). New records were collected following the same procedures outlined in *González et al. (2013)*, that is, from direct and indirect evidence of guanaco presence collected between the years 2000 and 2016 across several field campaigns. Indirect evidence of guanaco occurrence was assigned to a lineage by genetic and morphological evaluation of biological samples such as feces and dead tissues. Most of new records were collected in the northern section of the country in the Arica, Parinacota (i.e., 17°S latitude) and Coquimbo region (30°S). Each record was assigned to a 1 × 1 km cell defined by the resolution of the environmental datasets employed (see below). This resulted in a total of 298 records for *L.g. cacsilensis*, 837 for the mixed population, and 2,186 for *L.g. guanicoe.*

### Climate predictors

We limited the selection of environmental predictors to climatic variables (*Thuiller et al., 2006*; *Hu et al., 2015*). Similarly to what has been described in the literature (*Thuiller et al., 2004*), our previous work dismissed the importance of nonclimate predictors for guanaco distribution models in favor of exclusive climatic variables (*González et al., 2013*). We used all 19 bioclimatic variables from WorldClim (version 1.4) summarizing temperature and precipitation information worldwide (*Hijmans et al., 2005*). To reduce

collinearity, model overfitting, and the number of explanatory variables, we used a paired correlation analysis to inspect pairs of variables and removed variables with a large correlation coefficient (>0.8) (*Beaumont, Hughes & Poulsen, 2005*).

While the analysis was limited to the administrative bounds of Chile, all WorldClim variables were projected to UTM 19 South, with a one squared-kilometer of spatial resolution, spanning from latitudes 15° to 55°S and longitudes 60° to 80°W and a total area of 5,921,578 km$^2$ covering most of the Southern Cone.

## Future Climate Projections

The projection of future geographic distribution of niches was performed using the outputs of the Coupled Model Intercomparison Project 5 of the IPCC's methodology for the Fifth Assessment Report (AR5) (*Taylor, Stouffer & Meehl, 2012*). The two extreme GHG concentration scenarios, also known as representative concentration pathway (RCP), were used to project future climate niches. RCP2.6, the most optimistic scenario, considers a lower GHG concentration and projects average increases of temperature between 0.3° and 1.6 °C with 0.26–0.55 m increases of sea levels. RCP8.5, the most pessimistic scenario, considers higher GHG concentrations with a 2.6°–4.8 °C projected increase in mean global temperature and a 0.45–0.82 m rise of sea levels (*IPCC, 2013*). We chose both extreme scenarios to evaluate the minimum and maximum potential impact of climate change in the guanaco's distribution.

Given the large uncertainties of future climate predictions, the computing power availability and the exploratory nature of such models, we selected five general circulation models (GCM) among the 19 models used to generate the AR5. GCM's are physical climate models that simulate the interactive biophysical processes between the atmosphere, the ocean and the land (*Moss et al., 2010*). Selected climate models were: (1) CCSM4 model of the National Center of Atmospheric Research (*Gent et al., 2011*); (2) GFDL-CM3 model of the Geophysical Fluid Dynamics Laboratory (*Donner et al., 2011*); (3) GISS-E2-R model of the NASA Goddard Institute for Space Studies (*Nazarenko et al., 2015*); (4) HadGEM2-AO atmosphere model and (5) HadGEM2-ES earth system model, both of the Met Office Hadley Centre (*Collins et al., 2011*). Each scenario was evaluated for the short (2050) and medium term (2070).

## Niche modeling

Entropy maximization procedures in MaxEnt 3.3.3 k (*Phillips, Anderson & Schapire, 2006*) were used to model current and future geographic distributions of *L. guanicoe* and its lineages. MaxEnt uses a machine learning algorithm to generate predictions on the potential distribution of species based on their presence, pseudo-absences and a set of environmental variables. The software analyzes the multivariate distribution of environmental conditions of species occurrences to generate a spatially explicit probability map of lineage occurrence (*Franklin, 2009*). Such modeling approach has shown to have a good statistical performance compared to other types of modeling techniques (*Elith et al., 2006*) and is currently one of the most commonly used methods to understand habitat suitability,

niche structure, geographical species distribution (*Merow, Smith & Silander, 2013*) as well as to project environmental niches to future scenarios (*Hijmans & Graham, 2006*).

We performed 100 cross-validated replicates for each current and projected distribution model with logistic output, that unlike other outputs (i.e., raw and cumulative) assumes that a known observation probability can be assigned to each pixel and has thus been considered as a true approximation of presence (*Merow, Smith & Silander, 2013*). The "fade by clamping" option was used to avoid predictions beyond the observed geographical range during the training of the future distributions models (*Phillips, Anderson & Schapire, 2006*). All other parameters were kept at their default configuration (*Phillips, Dudík & Schapire, 2004*) as they have previously shown good performance in ungulate modeling (*Hu & Jiang, 2011*; *González et al., 2013*; *Hu et al., 2015*; *Quevedo et al., 2016*) and other taxonomic groups (*Phillips & Dudík, 2008*; *Fourcade et al., 2014*).

We used an ensemble forecasting framework to minimize the inherent variability introduced by the various forecast models employed, as proposed by *Araújo & New (2007)*. Therefore, we generated a model from the average of each bioclimatic variable produced by the five GCMs (i.e., $Bio1_{CC}$ + $Bio1_{GF}$ $Bio1_{GS}$ + $Bio1_{HD}$ + $Bio1_{HE}$), and then evaluated an average value for each variable from 100 replicates for both extreme RCP emission scenarios for the years 2050 and 2070. Hence, 400 projected guanaco distribution models were generated (i.e., 2 RCPs × 2 time frames × 100 replicates). The final results are four projected climate models for *L. guanicoe*, one for each RCP2.6 and RCP8.5 scenarios evaluated for years 2050 and 2070.

## Model evaluation, prediction, and spatial projection

Generated niche models were evaluated using a threshold-independent analysis of the area under the curve (AUC) provided by the receiver operator curve (*Phillips, Anderson & Schapire, 2006*; *Acevedo et al., 2010*; *Anderson & Raza, 2010*). These sensitivity tests model accuracy by calculating the proportion of true positives versus false positives. The resulting values range from 0 to 1, where model predictions are considered fair when obtained AUC values are above 0.7 (*Swets, 1988*; *Merow, Smith & Silander, 2013*). A 3:1 ratio was used to divide training and testing datasets (*Phillips, Anderson & Schapire, 2006*). AUC Jackknife analysis allowed to identify the contribution of each variable to final current and future models, and to allow the detection of those variables that significantly improve predictions for the occurrences of each lineage (*Phillips, Anderson & Schapire, 2006*).

We reclassified predicted habitat using a 0.25 threshold interval to label three habitat suitability classes: low suitability habitat when occurrence probability ranged between 25% and 50%; suitable habitat if occurrence probability was in the 50–75% interval; and high suitability habitat if occurrence probability was over 75%, values below 25% were considered as inappropriate habitat (*Hu & Jiang, 2012*; *González et al., 2013*; *Shrestha & Bawa, 2014*).

## Changes in distribution surface and incidence in protected areas

The areal extent for each suitability class predicted by each model, current and projected, were compared to determine habitat loss (or gains) under the various climate change

scenarios evaluated. We used the software BioSARN v.1 to calculate the amount of area gained or lost and to estimate differences between models (*Heap, 2016*). These results were classified into three categories: (a) Areal loss, when future prediction show a decrease of the areal extension compared to current niche models; (b) Areal gain, produced when future prediction add area to current niche models; (c) Unchanged areas, when climate change predictions show no impact on current guanaco's distribution.

In addition, the fraction of future distribution covered by the system of PA in Chile was estimated. All categories offering some level of protection were considered: national parks, national reserves, biosphere parks, national monuments, national patrimony, and private PA as of 2011. RAMSAR sites (as of 2012) were also included as they constitute the most important feeding grounds for guanaco in the hyper-arid north of Chile (*Squeo et al., 2006*).

## Evaluation of PNC or PND

Phylogenetic niche conservatism and phylogenetic niche divergence among lineages and the mixed population were evaluated through their current and projected niches for the most extreme scenario (i.e., RCP8.5) in 2070 using ENMTools v.1.4.3 (*Warren, Glor & Turelli, 2010*). Niche overlap between lineages was calculated with the statistical indices "*I*" (derivative of Hellinger's distance) and "*D*" (Shöener's D) which may take values going from 0 (i.e., no overlap) to 1 (i.e., full overlap between lineages' niches). We used the Identity test to evaluate the hypothesis of niche equivalence (*Graham et al., 2004*) and the Background similarity test to evaluate the hypothesis of niche similarity (*Peterson, Soberón & Sánchez-Cordero, 1999*). The Identity test quantitatively assesses whether the niche space for two compared lineages are equivalent by comparing the actual niche to a null niche model generated from a randomized pool of locations for each lineage. This allows to effectively evaluate whether niche spaces are equal, under the premise that, if they are, they should be able to predict each other (*Warren, Glor & Turelli, 2010*). Because the Identity test strongly depends on accurate representations of species habitat suitability, it is known to be sensitive to the particular sampling scheme employed, and therefore less suitable to compare allopatric niches (*Warren, Glor & Turelli, 2010*). The Background similarity test compares the niche difference between allopatric lineages by contrasting the niche of a "focal" lineage to the niche built from the background locations of a second lineage. If there is similarity between these, the null model should predict the niche of the second lineage. We repeated each test 100 times to produce a simulated distribution of *I* and *D* values and to evaluate significance using a threshold of 0.1 (two-tailed for background similarity test, and one-tail for identity test) (*Warren, Glor & Turelli, 2010*; *Guisan, Thuiller & Zimmermann, 2017*). We considered outcomes as indicative of PNC between lineages when observed *I* and *D* values fell within the simulated distribution. On the other hand, when the observed values fell outside of the simulated distribution, they were assumed to be indicative of PND between lineages.

## RESULTS

### Selection of climate variables and current distribution model

After removing correlated variables, the final subset of independent bioclimatic variables used in this analysis was composed of: annual mean temperature (Bio1), temperature seasonality (Bio4), annual temperature range (Bio7), annual precipitation (Bio12), and precipitation seasonality (Bio15). See correlation analysis in supplemental Fig. S1.

The major contribution to the current distribution of *L. guanicoe* was given by the annual range of temperature (28.2%, AUC = 0.84), whereas *L.g. cacsilensis* was dominated by precipitation seasonality (66.7%, AUC = 0.95), the mixed population by annual precipitation (36.7%, AUC = 0.93), and the southernmost *L.g. guanicoe* was mostly driven by annual mean temperature (35.8%, AUC = 0.91). Detailed Jackknife's analysis can be seen in supplemental information, Fig. S2.

All current distribution models generated for *L. guanicoe* and its lineages, presented a good performance with mean AUC values over 0.89. The resulting geographic range for guanaco spanned for about a third of the Chilean continental surface. The geographic areas covered by *L.g. cacsilensis* and *L.g. guanicoe* were of 47,148 and 100,539 km$^2$, respectively. The mixed population showed a geographic extent of 84,976 km$^2$. Interestingly, our models had a 20% difference when comparing areas from the sum of lineages modeled independently and the total area modeled with all the lineages pooled as if they were a single lineage (232,664 vs. 284,499 km$^2$, respectively). Full maps and predictions are available in supplemental information Fig. S3.

### Projected distribution models

As for current distribution models, the sensitivity analysis yielded a large mean AUC >0.9. After suitability categorization (Fig. 1), our results show that while the geographical distribution pattern of guanaco is conserved, quantitative assessment of the distribution surface reveals a downward trend in both scenarios of climate change for 2050 and 2070 (Table 1).

Marginal decreases of habitat suitability is observed under the RCP2.6 scenario for the years 2050 and 2070 (259,577 vs. 254,979 km$^2$, respectively). The small areal reduction under both models is only of 6% and 7.6% of the current area. However, this decrease is more pronounced under the RCP8.5 scenarios in which a 13% and 20.7% reduction is quantified for the years 2050 and 2070, respectively (i.e., 240,505 and 218,841 km$^2$).

### Surface change between current and projected distribution models

Projected distribution models under the more optimistic scenario showed an increase of high quality habitat and a net loss of medium and low quality habitat, while the projections under the worst scenario indicate a generalized decrease in habitat suitability. Although no important loss of potential distribution areas are apparent, a large decrease
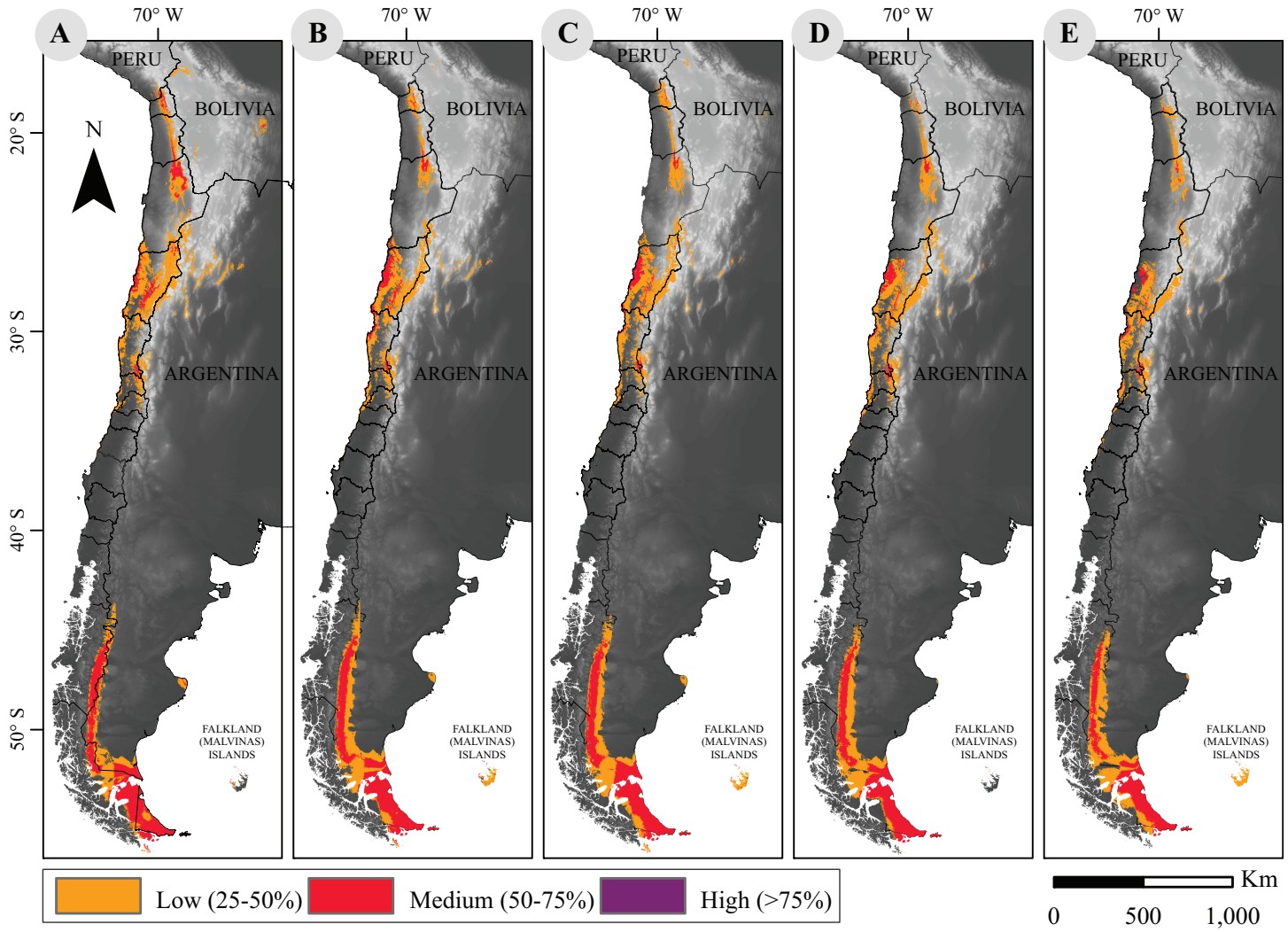

**Figure 1 Current and projected distribution model of guanaco lineages in South America.** (A) Current distribution; and projected distributions (B) under RCP2.6 to 2050; (C) under RCP2.6 to 2070; (D) under RCP8.5 to 2050; (E) under RCP8.5 to 2070. These surfaces were classified according to habitat suitability. Base Map Elevation Data: CIAT-CSI SRTM (http://srtm.csi.cgiar.org).

of areas with high quality habitat may occur under the worst climate change scenario evaluated (Table 1).

Surface losses and gains under future climate projections are described in Table 2. Both scenarios showed that a large fraction of the guanaco range will remain unchanged. The more optimistic projection (i.e., RCP2.6) indicated an average loss of 67,042 km$^2$ between 2050 and 2070, and a niche displacement (i.e., gain) of 48,225 km$^2$ on average between such time periods. A reversed trend was observed under the more pessimistic scenario (i.e., RCP8.5) with the larger change predicted for 2070. Such prediction forecasts a reduction in guanaco's niche by 37%, with a surface loss of 103,367 km$^2$ and a geographic distribution of 172,786 km$^2$. Likewise, the smallest niche displacement was observed for this period, with 46,089 km$^2$ of areal gain (Fig. 2).

**Table 1 Geographic distribution area (km²) of current potential distribution of *L. guanicoe* across habitat suitability categories.**

| | | RCP2.6 | | RCP 8.5 | |
|---|---|---|---|---|---|
| Suitability category | Current (Fig. 1A) | 2050 (Fig. 1B) | 2070 (Fig. 1C) | 2050 (Fig. 1D) | 2070 (Fig. 1E) |
| High (>75%) | 23 | 71 | 258 | 3 | 0 |
| Medium (50–75%) | 102,693 | 94,574 | 88,295 | 80,653 | 66,344 |
| Low (25–50%) | 173,353 | 164,932 | 166,427 | 159,849 | 152,497 |
| Total | 276,069 | 259,577 | 254,979 | 240,505 | 218,841 |

Notes:
Environmental niche models are projected to years 2050 and 2070 under the most extreme greenhouse gas emission scenarios. RCP2.6 represents climate model under the less severe emission scenario and RCP8.5 the scenario under the largest greenhouse gas emission.

**Table 2 Percent change of distribution area between current and projected models for 2050 and 2070 under the most extreme climate change projections.**

| | RCP2.6 | | RCP8.5 | |
|---|---|---|---|---|
| | 2050 (Fig. 2A) | 2070 (Fig. 2B) | 2050 (Fig. 2C) | 2070 (Fig. 2D) |
| Losses (km²) | 66,634 (24%) | 67,450 (24%) | 86,540 (31%) | 103,367 (37%) |
| Unchanged (km²) | 209,519 (76%) | 208,703 (76%) | 189,613 (69%) | 172,786 (63%) |
| Gains (km²) | 50,106 (18%) | 46,343 (17%) | 50,968 (18%) | 46,089 (17%) |

Notes:
Areal losses, unchanged and gains in square kilometers for each RCP model with respect to current potential distribution of guanaco. Percentage changes are shown in parentheses.

## Evaluation of PNC and PND

Niche overlap and equivalence tests showed a large overlap between the contact population and *L.g. cacsilensis* for niche models under current and worst scenarios (Table 3). While limited niche overlap was reported, the overlap between the northern lineage and the mixed population increased under future climate change scenarios. On the other hand, the overlap between current and projected niches for *L.g. guanicoe* with the other groups was smaller, particularly with the northern lineage (*L.g. cacsilensis*). When using the results of this latter analysis as "empirical values" to perform identity and background similarity tests (Tables 3 and 4), we were able to show that statistical differences existed when comparing current niches and projected niches. This indicates that the climatic requirements between the lineages and mixed population are not equivalent.

Background similarity test showed that *L.g. cacsilensis* share climatic similarities with the mixed population (Table 4). However, such similarity is not reciprocal when comparing the climatic requirements of the mixed population to those of the northern lineage—a possibility mentioned in *Warren, Glor & Turelli (2010)*. A similar situation occurred when comparing projected niches under the worst climate change scenario (RCP8.5), where results suggested that *L.g. guanicoe´s* niche will closely resemble the future climatic niche of the mixed population, in spite of not sharing any current similarity.

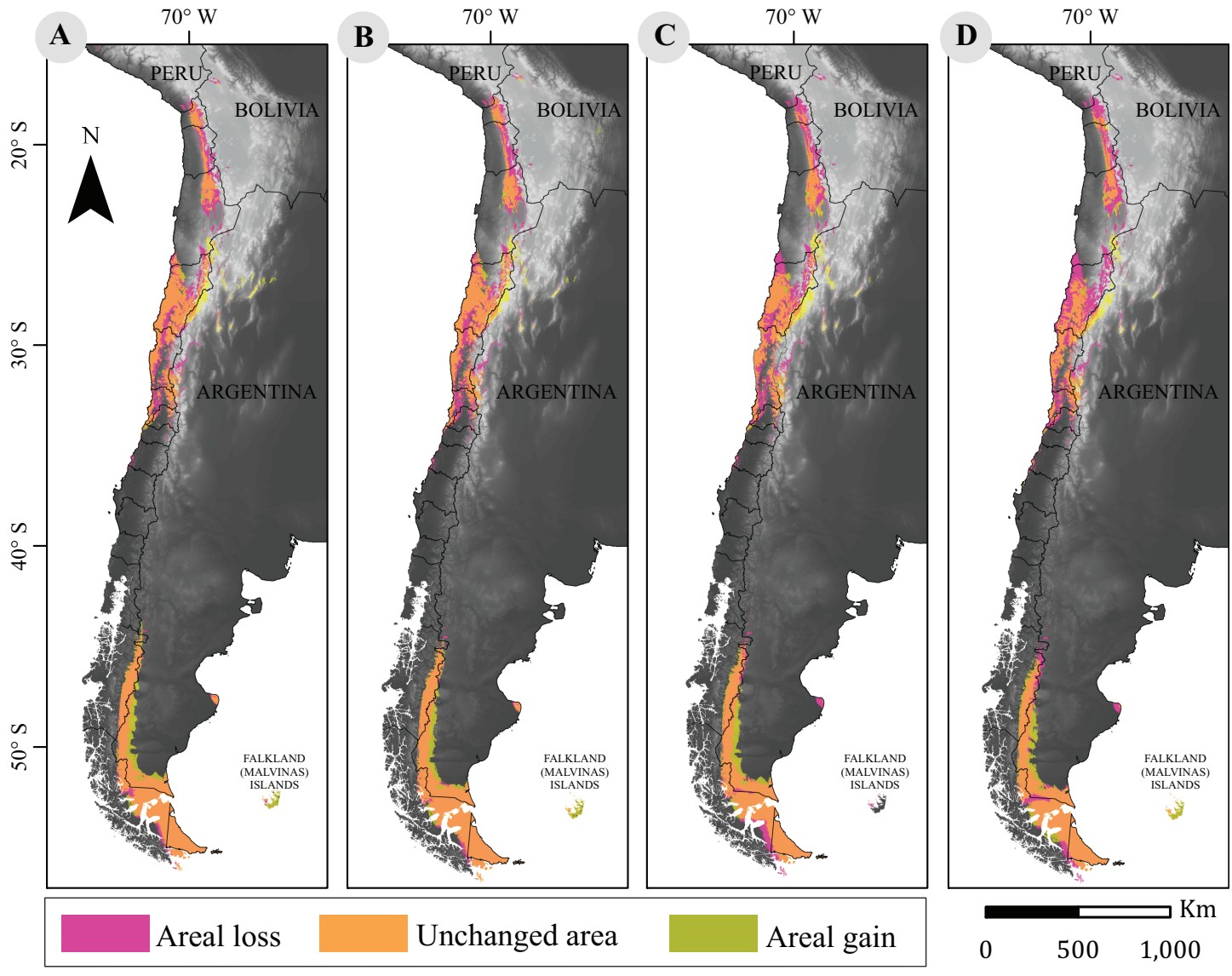

**Figure 2 Cartographic projection of changes in distribution.** Pink surface corresponds to areal loss and representing the areal fraction of the climatic niche that exists only in the current model. Olive surface corresponds to areal gains and represents the climatic niche area that exists only in the projected model. Orange surface corresponds to the area that has remained unchanged and represents the surface of the climatic niche that exists in both models (current and projected models). Panel (A) is the comparison between current and projected model under RCP2.6 to 2050; (B) the comparison between current and projected model under RCP2.6 to 2070; (C) the Comparison between current and projected model under RCP8.5 to 2050; (D) is the comparison between current and projected model under RCP8.5 to 2070. Base Map Elevation Data: CIAT-CSI SRTM (http://srtm.csi.cgiar.org).

The remaining comparisons between climatic niches, current and future, showed that the climatic requirements for each of the lineages analyzed are significantly different (Table 4).

## Projected distribution models and conservation in Chile

The current network of PA in Chile covers a vast area of approximately 256,550 km$^2$, according to 2016 data, and shows a limited overlap with modeled guanaco distribution.

**Table 3 Niche identity test.**

| Niche model | Compared lineages | Empirical value (Niche overlap) | | Identity test (Niche equivalence) | |
|---|---|---|---|---|---|
| | | *I* | *D* | *I* | *D* |
| Current | *L. g. cacsilensis*—Mixed population | **0.283** | **0.089** | 0.923 | 0.686 |
| | *L. g. guanicoe*—Mixed population | 0.178 | 0.058 | 0.922 | 0.711 |
| | *L. g. cacsilensis*—*L. g. guanicoe* | 0.133 | 0.033 | 0.922 | 0.596 |
| Projected (2070) | *L. g. cacsilensis*—Mixed population | **0.471** | **0.208** | 0.930 | 0.760 |
| | *L. g. guanicoe*—Mixed population | 0.135 | 0.039 | 0.950 | 0.800 |
| | *L. g. cacsilensis*—*L. g. guanicoe* | 0.090 | 0.015 | 0.890 | 0.630 |

**Notes:**

These results correspond to the comparison between the empirical values (niche overlap) and values of percentiles 0.1 of the null distribution (one tailed, *Warren, Glor & Turelli, 2010*). The "*I*" and "*D*" statistics allow to compare the overlap between the replicas of this test. If the empirical value is within the range of values observed in the percentages, the hypothesis of niche equivalence is supported (shown in bold type).

**Table 4 Background similarity test.**

| | Compared lineages | | Niche overlap | | Background similarity test | | | |
|---|---|---|---|---|---|---|---|---|
| Niche | | | | | *p* = 0.01 | | *p* = 0.90 | |
| Model | Focal | Background | *D* | *I* | *D* | *I* | *D* | *I* |
| Current | *L. g. cacsilensis* | Mixed population | **0.089** | **0.283** | **0.082** | **0.260** | **0.115** | **0.320** |
| | Mixed population | *L. g. cacsilensis* | 0.089 | 0.283 | 0.094 | 0.304 | 0.108 | 0.336 |
| | *L. g. guanicoe* | Mixed population | 0.058 | 0.133 | 0.136 | 0.358 | 0.140 | 0.366 |
| | Mixed population | *L. g. guanicoe* | 0.058 | 0.133 | 0.091 | 0.302 | 0.107 | 0.332 |
| | *L. g. cacsilensis* | *L. g. guanicoe* | 0.033 | 0.178 | 0.091 | 0.273 | 0.124 | 0.332 |
| | *L. g. guanicoe* | *L. g. cacsilensis* | 0.033 | 0.178 | 0.136 | 0.359 | 0.141 | 0.368 |
| Projected (2070) | *L. g. cacsilensis* | Mixed population | 0.208 | 0.471 | 0.158 | 0.387 | 0.169 | 0.410 |
| | Mixed population | *L. g. cacsilensis* | 0.208 | 0.471 | 0.055 | 0.211 | 0.074 | 0.251 |
| | *L. g. guanicoe* | Mixed population | **0.039** | **0.135** | **0.033** | **0.123** | **0.042** | **0.148** |
| | Mixed population | *L. g. guanicoe* | 0.039 | 0.135 | 0.011 | 0.055 | 0.016 | 0.072 |
| | *L. g. cacsilensis* | *L. g. guanicoe* | 0.015 | 0.090 | 0.006 | 0.042 | 0.010 | 0.058 |
| | *L. g. guanicoe* | *L. g. cacsilensis* | 0.015 | 0.090 | 0.018 | 0.080 | 0.020 | 0.085 |

**Notes:**

The first column indicates whether the analysis was applied on current or projected niche models. Second and third columns indicate compared lineages and focal lineage used for the comparison. These results show the comparison between the empirical values (niche overlap results) and 0.1 and 0.9 percentiles of the null distribution (two tailed, *Warren, Glor & Turelli, 2010*) delivered by the test. If the empirical value is within the range of values observed in the percentages, the hypothesis of niche similarity is supported (shown in bold type).

In fact, our analysis shows that a 9.8% (i.e., 19,402 km$^2$) of the species current distribution overlaps with a PA. When looking at projected distributions for 2070, our results showed that such overlap will decrease to 6.2% (15,772 km$^2$) under the best scenario (RCP2.6). Similarly, under the worst scenario (RCP8.5), the overlap will be of 5.7% (12,434 km$^2$) (Fig. 3).

## DISCUSSION

Understanding species' response to climate change is crucial in order to adequately manage conservation efforts (*Thomas et al., 2004*; *Araujo & Rahbek, 2006*; *Warren et al., 2013*). Several authors have already warned about the dire consequences of climate change on

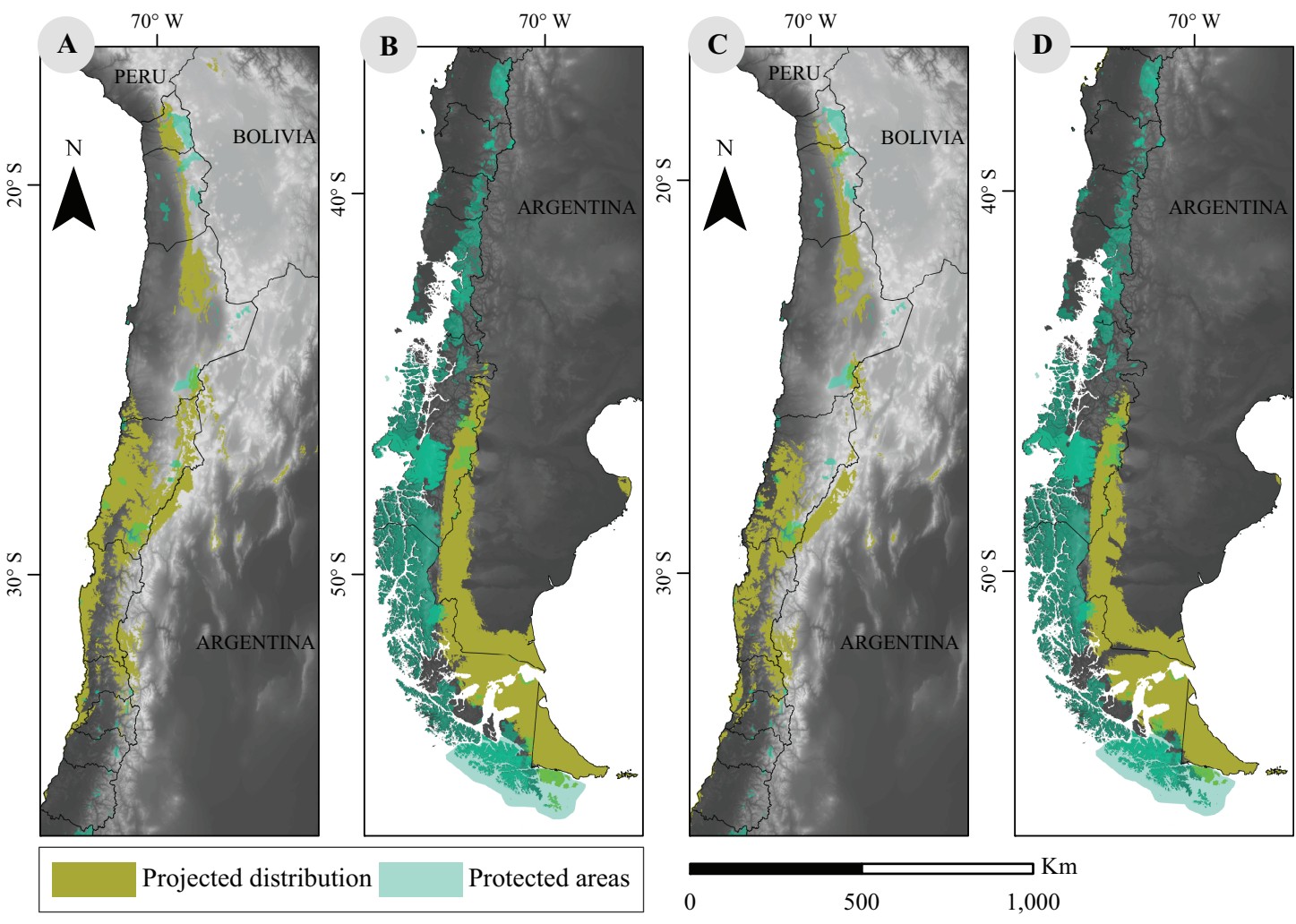

**Figure 3 Overlap between projected environmental niche models under two extreme climate scenarios and distribution of protected areas in Chile.** (A) and (B) are projections based on RCP2.6 scenarios; (C) and (D) correspond to RCP8.5 scenarios. Protected categories consider National Parks, National Reserves, Biosphere Reserves, National Monuments, Protected National Heritage, RAMSAR sites (2012), and Private Protected Areas (2011) (*IDE, 2016*). Base Map Elevation Data: CIAT-CSI SRTM (http://srtm.csi.cgiar.org).     

ecosystems and across a wide range of taxa (*Walther et al., 2002*; *Quintero & Wiens, 2013*; *Muñoz-Mendoza et al., 2017*). Our worst-case scenario analysis suggests that the guanaco will have lost up to a fifth (~21%) of its current geographic distribution by 2070. Although predicted changes will barely be noticeable, the classification and quantification of suitable habitat provided substantial insights on the vulnerability of this species to climate change showing that such changes will additionally result in a confinement to suboptimal quality habitats supporting general trends outlined in the literature (*Urban, 2010*; *Pecl et al., 2017*). In particular, our quantification of areal changes between current and future distribution under the worst-case scenario, suggested an average loss of 34%, compared to 17% of areal gains (i.e., new distribution areas available) between 2050 and 2070.

Our analysis strongly associated the northern lineage *L.g. cacsilensis* to precipitation seasonality and predicted a southward shift in the future distribution of guanaco. However, our model outputs essentially predicted an eastern distributional shift into areas of low suitability between the Arica and Parinacota (18°S), and the Atacama regions (27°S) (Fig. 1). Other authors have predicted similar changes in the distribution area of ungulates in arid and semiarid environments as, for example, the silver dik-dik (*Madoqua piacentinii*, a small antelope), in the southeastern coast of Somalia (*Thuiller et al., 2006*). In other cases, extinction risks have increased with the intensification of drought episodes, as it has been seen for the hartebeest (*Alcelaphus buselaphus*, an African antelope), and the waterbuck (*Kobus ellipsiprymnus*, a large sub-Saharan antelope) (*Duncan et al., 2012*).

In the case of the guanaco in Chile, future distribution modeled here not only shows a confinement to specific zones, such as coastal and central valleys in the Atacama region (27°S), and coastal and Andes Mountains between Coquimbo (29°S) and Valparaíso (33°S) regions, but also a reduction of the distribution extent of the mixed population. This roughly coincides with a predicted 5–15% rainfall reduction for the next decades between the Copiapó River (27°S) and the Aysén river basin (47°S) (*Rojas, 2012*; *Garreaud et al., 2017*).

In order to better understand spatial dynamics of guanaco's populations across their distribution range, we seeked to address whether niche conservatism or divergence prevailed under different climate change regimes. The basic assumption is that niches are diagnostic traits that help us understanding how species deal with climate-induced changes in their habitat (*Wiens & Graham, 2005*; *Alvarado-Serrano & Knowles, 2014*). For instance, sister lineages should most likely exhibit closely similar niches, and point towards PNC (*Webb et al., 2002*; *Wiens & Graham, 2005*; *Losos, 2008*; *Warren, Glor & Turelli, 2008*). Therefore, we expected to find niche similarity (i.e., PNC) between the niches of guanaco lineages. Nevertheless, we found stronger evidence for PND among *L.g. cacsilensis* and *L.g. guanicoe*, hence supporting the existence of only two ESUs for the guanaco, as proposed by *Marin et al. (2013)*: one lineage in the northwest represented by *L.g. cacsilensis* and another in the southeast represented by *L.g. guanicoe* (see Tables 3 and 4). Two interesting results emerged when lineages' niches were compared with the mixed population: (i) the current niche of northern *L.g. cacsilensis* is similar to the current niche of the mixed population; (ii) the future niche of the southern *L.g. guanicoe* is projected to include, and resemble, the future niche of the mixed population. Nevertheless, the current niche of the mixed population does not share any statistically significant similarity with the northern nor the southern lineages (see background similarity test results in Table 4). This suggests that *L.g. cacsilensis* is, given its actual climatic requirements, more likely to adapt to current climatic conditions across the mixed population's habitat, pointing towards the existence of PNC processes. Whereas when we look at projected climate conditions, *L.g. guanicoe* will most likely experience a future expansion of its environmental niche towards the projected mixed population's habitat. Conversely, the mixed population will continue to limit its distribution to the small and restricted areas of northern Chile, in spite of its recent dispersal history through extant barriers (*Marin et al., 2013*). In summary, PNC seems to be more important when

current niches are analyzed, while PND emerges as the important process under future projections of climate change.

The high genetic variation observed for the guanaco lineage in Patagonia indicates that this area may have functioned as a climatic refuge for the species (*Fuentes & Jaksic, 1979*; *González et al., 2013*). In order to safeguard the evolutionary potential of the species, conservation efforts should take into consideration the projected distributions of guanaco lineages (*Pecl et al., 2017*).

From our results, and under the best scenario, the current location of PA will only decrease a 3% of guanaco's protected distribution range, but will shift to lower quality habitat, as discussed above. Projected distributions models proposed in this study are by no means a prognosis of the fate of guanacos in Chile, as they outline the distribution probabilities based on possible scenarios given the future GHG emissions (*IPCC, 2013*). Furthermore, the limited geographic locations that fed our models and the uncertainties associated with GCMs (*Buisson et al., 2010*) have likely permeated our predictions (*Moss et al., 2010*). In fact, the net effect of our conservative forecasts may underestimate current and future species distribution both within the boundaries of Chile and beyond. For instance, the species may be influenced by a range of environmental conditions outside Chile that are currently not represented among the presence localities in Chile. Thus, when projecting the future distribution within Chile, these areas may appear as not environmentally suited for the presence of the species, altering the correct estimation for the area (or percentage) of range contraction and the area (or percentage) of the range in PA. Furthermore, addressing congruence between restricted and broad scale predictions, *Titeux et al. (2017)* suggested that local models might omit the warmest and coldest parts of future distribution, projecting a larger decrease in future species richness at warmer temperatures. Therefore, the validity of the proposed distribution of guanaco's future range in Chile likely depends on the consistency between the environmental niche used and the values present elsewhere, as well as on the extension of guanaco's presence in extreme temperatures areas. Understanding how these observations could translate to guanaco's species and lineage distribution will certainly contribute to the ongoing research efforts currently underway in Argentina and Chile. Hence, it is likely that efforts to expand the sampling dataset to consider guanaco's full range will also increase its climate niche definition and improve the predictions of the future distribution of the species. The work presented here represents a conservative view of guanaco's range that allows the evaluation of the evolutionary aspect of niche conservatism hypotheses based on the best knowledge of the species natural history.

## ACKNOWLEDGEMENTS

The authors would like to thank PeerJ editor Bruno Marino and two anonymous reviewers for their suggestions, which contributed significantly to improve the original manuscript.

### Funding

This work was supported by the government of Chile and CONICYT through grants Fondecyt # 1161280, FONDEF # D10I1038 to H. Samaniego and Doctoral Fellowship # 21150634 to D. Alò. The funders had no role in study design, data collection and analysis, decision to publish, or preparation of the manuscript.

### Grant Disclosures

The following grant information was disclosed by the authors:
Government of Chile: Fondecyt # 1161280.
CONICYT: FONDEF # D10I1038.
Doctoral Fellowship: # 21150634.

### Competing Interests

The authors declare that they have no competing interests.

### Author Contributions

- Andrea G. Castillo performed the experiments, analyzed the data, prepared figures and/or tables, authored first draft, and approved the final draft.
- Dominique Alò authored or reviewed drafts of the paper, approved the final draft.
- Benito A. González contributed reagents/materials/analysis tools, authored or reviewed drafts of the paper, approved the final draft.
- Horacio Samaniego conceived and designed the experiments, performed the experiments, analyzed the data, contributed reagents/materials/analysis tools, authored or reviewed drafts of the paper, approved the final draft.

### Data Availability

The raw data are provided in the Supplemental Files.

### Supplemental Information

Supplemental information for this article can be found online at http://dx.doi.org/10.7717/peerj.4907#supplemental-information.

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
