# Peer review of "Change of niche in guanaco (Lama guanicoe): the effects of climate change on habitat suitability and lineage conservatism in Chile"

_PeerJ, doi:10.7717/peerj.4907_

## Round 0.1 · original submission · Major Revisions

This manuscript warrants major revision as the data presented are of a unique perspective and should be published. However, a number of suggestions have been noted by the reviewers that if addressed may improve the study for publication in PeerJ. Please note that one major criticism is that as only Chilean distribution of Lama guanicoe is analysed, only data from Chile should be used in the analysis. In addition, attribution of the raw data appears to be problematic based on the coordinates of the data provided (they are displaced to the east when mapped in a GIS, using the UTM 19S coordinate system). Please address this concern upon re-submission.

Reviewer 1 ·

Basic reporting

Overall, the manuscript is well-written, the use of professional English throughout is clear and unambiguous (although some minor edits are indicated below), and conforms to professional standards of expression. The introduction provides sufficient background to the study. The literature references are up to date and sufficient.
The structure of the article conforms to the acceptable format. The figures are relevant to the content of the article (see below for suggestions regarding the manuscript's figures). The raw data has been made available, however, there seems to be a problem with the coordinates of the data provided (they are displaced to the east when mapped in a GIS, using the UTM 19S coordinate system).

Content edits
lines 129-131. The statement in the sentence starting with "For instance,..." does not seem correct. A generalist species tends to be more adaptable in the face of climate change. It is usually species with narrow or more specific habitat requirements and/or with low tolerance to environmental conditions that are more sensitive to climate change.

lines 394-395. In the sentence that starts with "Our worst-case scenario..." the authors should clarify that the percentage lost is for the Chilean range only.

Caption for Figure 1. I do not think it is necessary to explain all four software programs used in the analyses, as it was already explained in the main text.

Figures 2-4. It appears that the country labels in the maps could fit horizontally, which is visually better. The label for the Falkland Islands does not seem to be necessary, but if the authors decide to include it, it should be in a smaller font than the country labels. Also, in most maps it is shown as Falkland (Malvinas) Islands, as it is a disputed territory.

Minor edits
line 78. Replace "South American cone" with "southern cone"
lines 88-89. The sentence beginning with "In spite of this,..." does not actually contrast the statement in the previous sentence.
line 95. "West" should be in lower case
line 100. "Northwestern" and "Southwestern" should be in lower case
lines 116-117. should say "growth of vegetation"
line 121. Replace "discusses" by "discussed"
line 122. "..., as has been discussed elsewhere for other species." Should include citations.
lines 125-129. The sentence beginning with "However,..." does not actually contrast the statement in the previous sentence.
line 166: Should read "Species" instead of "Specie"
lines 178-179. "In time of future anthropogenic landscape transformations (citations)" is not a complete sentence.
line 182. All descriptions are in past tense, so change "We use..." for "We used..."
line 327. Replace "show" by "has"
line 363. Replace "worse" by "worst"
line 460. Delete "a" in "a 11% of the future guanaco distribution"
Captions for Tables 2 and 3. Replace "de" by "the" and "sever" by "severe"
Caption for Table 3. No need to explain the analysis here, as it was already explained in the main text. The caption can start with "The results in the table correspond...". Delete last sentence. Instead, consider using "If the empirical value is within the range of values observed in the percentages, the hypothesis of niche equivalence is supported (shown in bold type)."
Caption for Table 4. Delete last sentence. Instead, consider using "If the empirical value is within the range of values observed in the percentages, the hypothesis of niche similarity is supported (shown in bold type)."

Experimental design

The manuscript is within the scope of the journal and the research questions are well-defined. The methods are described in sufficient detail to be replicated by other researchers. However, my main concern with the study is the regional scope of the analysis (at the country level). The guanaco's distribution includes the southern half and north west of Argentina, south of Bolivia, and southwestern Peru. Therefore, restricting the analysis to Chile may have implications for capturing the full environmental range for the species. There are several studies addressing this issue (e.g. Pearson and Dawson 2003 GEB 12:361-372; Thuiller et al. 2004 Ecography 27:165-172; Barbet-Massin et al. 2010 Ecography 33:878-886; Titeux et al. 2017 DOI: 10.1111/ddi.12634). From Thuieller et al's (2004) main conclusions: "Using restricted data (similar to not capture the full species’ environmental range) reduces strongly the combinations of environmental conditions under which the models are calibrated, and reduces the applicability of the model for predictive purposes (Pearson and Dawson 2003). This problem has important implications when future projections of species distributions are sought. In particular, species niche from restricted data sets might be seen as analogous to the modelling of species niche from a limited geographic location not covering the complete range of environmental conditions in which species may occur." It would be interesting to see how the projected distribution of the guanaco in Chile may change if the entire environmental range is considered.
Related to the point above, looking at Figure 3, the larger areal gains in the different climate change scenarios are in southern Argentina, where the species already occurs.

Minor comment.
How were the CMIP5 variables processed to match the WorldClim variables?

Validity of the findings

As mentioned in the previous section, my main concern with the study is the generation of models with geographically restricted data. Using these models to predict distribution shifts under climate change may produce biased projections. Therefore, the validity of the findings may be questionable.

Additional comments

I suggest the authors redo the analysis including the entire range of the species.

Reviewer 2 ·

Basic reporting

This is a potentially interesting work on current and forecasted distributions of guanaco Lama guanicoe in Chile, which could merits publication, but only after substantial changes and clarifications. The mss models both present day and forecasted distribution under different climate change scenarios, and a somewhat confusing section about niche divergence between species’ lineages. Besides, it includes some information about changes in protected distribution under the mentioned CC scenarios. The mss has some interesting results, although the mixture of hypothesis and analytical approaches makes its understanding somewhat difficult. I congratulate the authors for their extensive data set and the effort made in using modern techniques to analyse it.

The manuscript needs a deep English revision. Writing is in general rather wordy, and the text is oversized and confusing in some parts. The paper therefore would benefit from shortening and improving writing quality. Another problem is that, in general, writing tends to suggest that causality is demonstrated, but the authors should be careful about that, because the data presented here are correlational and results are strongly dependent on original variables used for the modelling. Another main problem of the paper is that statistical and modelling procedures are unclear.

Introduction
Introduction is rather long and worthy, as other sections of the mss, and should be substantially reduced. The main idea of the paper is rather hidden and I had difficulties to focus the paper. Some shifts of scale and conceptual jumps are found within and between paragraphs respectively. Objectives are not well connected with predictions. Especially, I found some inconsistencies in the relevance afterwards gave to some hypotheses (PNC and PND) and the little interest and justification given to them in the Introduction. Some recent references are needed, especially in the context of guanaco’ niche (see below).
I am especially concerned with the lack of clear predictions for these supposed phylogenetic relationships between lineages (see below).

References
Some basic and recent references are lacking. I suggest making some search, as I have done by myself, on guanaco and niche. See for instance: Traba et al., 2017. Oikos 126, Iranzo et al. 2013, Plos One.

Structure
Significant improvements have to be applied to fit the standards of PeerJ, both in relation to citing references and in figures and tables.

Figures
The mss has an inflation of figures. I suggest decreasing the number of figures. I have had some difficult in understand several of them, especially maps when colour print is lacking. Besides, several figures have typographic mistakes (see below). I suggest a thorough revision to easily amend these problems.

Raw data are supplied. However, some inconsistencies between maps and main text can be found, as the text is circumscribed to Chile, but maps show areas in Argentina and other countries. Please, authors should clarify this, as a main problem with original raw data may cause bias in modelling outputs (see below).

Experimental design

The paper is novel to my knowledge, and its findings could improve basic and theoretical information on guanaco, though it could be of special interest only for specialist.

Workflow is rather difficult to follow, and has some typographic mistakes. After reading the main text, I find this figure unnecessary.

One of my main concerns is related to raw data used for modelling present day distribution. As only Chilean distribution is analysed, only data from Chile should be used (but see my comment about Figures and maps). However, modelling only-climate distribution of a widely distributed ungulate with just a data subset may provoke a biased result about the climatic niche of the species. This is paradigmatic when analysing results from MaxEnt modelling, and this was made with just a small bunch of climatic variables, being all of them pretty relevant to explain the distribution of the three response variables: the two lineages and the species as a whole.

This is especially important when analysing phylogenetic relationships between lineages (which are not yet even subspecies, to my knowledge) of a potentially highly mobile species, and with a wide South American distribution. If these predicted phylogenetic differences (though lacking a direction of change) come from an evolutionary perspective, and the authors are not exclusively seeing the ecological output of a partial response to just a few predictive factors, they could not expect such changes on ecological time. As the distribution model is restricted to just the very same conditions of the present occurrences, and the authors do not consider all the many other locations of guanaco throughout the continent, they may perfectly be observing the result of local contingent plasticity, forced by historical processes. Including the whole distribution in the modelling could have driven to higher variability in input variables. I suggest to avoid this PNC/PND section in the mss, which could be exclusively focused on modelling present and forecasted distribution.

Regarding statistics, I have some concerns in relation to the myriad of stats used. I hardly follow all the methods, as clear research questions linked to methods are not so evident. Obviously, MaxEnt results are highly dependant on input variables, as AUC increases as correlation between variables does. As I have already mentioned my concerns on raw data, I have also some doubts about variables used in the modelling. As the authors have used only-climate variables, just on a subset of the spatial distribution, inconclusive and biased results may emerge. For instance, high similarity between lineages in relevance of climate variables is found. I suggest to include land-use / land-cover variables to improve the models and correctly asses the niche volume of the species (and its lineages). I also suggest to use other alternative methods to test niche overlap and segregation. I have some doubts about other packages, as Biosarn, which is not adequately explained. I have some doubts about two-tailed test (L304; see specific comments).

Validity of the findings

Discussion is in general rather speculative, and writing tends to give the impression that causality is demonstrated. Conclusions are vague and undefined, and deserve to be deleted.

Additional comments

Specific comments

Abstract

In general, I find the abstract confusing, though this could due to the necessity of improving the English. First sentence (L22-24) is unnecessary.
L26-28. Related to climate Change? Two very different objectives in the same mss.
L29 and ss. No information on data and methods is provided in the abstract. But information on the species is, what is already mentioned in the introduction.
L33-34. Contact population is inconcrete for the abstract.
L34. Prescribe?
L34-41. Authors should make an effort in shortening and clarifying these results.
L41. Lineages names should go above (L-27).
L43. Not necessary for an abstract.
L50. What is contact population niche?
L51-52. Final sentence is inconcrete and deserves deleting.

Introduction

L54-56. Unnecessary.
L56-61. Authors could consider merging but reducing here.
L61-68. Writing is wordy and confusing, mixing conceptual scales.
L75. Is Magellan the same than Magallanes (L73)? Please, clarify
L80. Roaming? May the authors want to say remaining?
L82-83. All the species are characterized by this.
L86. Is really necessary this?
L87. In spite of this? Maybe the authors wanted to say however?
L89. These qualities?
L91. Please, change feeds scavengers by is the prey of…
L94. Authors should homogenise the use of capitol or lowercase letters throughout the text. Please, see the standards of PeerJ.
L97. References are lacking here.
L105 and ss. All this about contact population is unclear. Does it have individuals from both lineages? How do you know that? The distribution was predicted to occur? But is it or not? How is the relation with predictors?
L115. Please, see standards of PeerJ for citing references.
L116. Growth of
L122. References are lacking here.
L122 and ss. Overwriting, please simplify.
L125. However?
L126. Impressive? I don’t understand what the authors want to say here.
L127-132. It needs deep revision. Confusing and oversized.
L133-137. Already mentioned.
L137 and ss. Already mentioned.
L140. Is difficult to ascertain if this contact population does exist or is only predicted.
L141. This asseveration is not substantiated.
L144. I hardly share this asseveration.
L145 and ss. This argumentation is out of focus of the mss. Maybe it deserves another mss at all.
L153. Is ESUs the same than lineages?
L157. surface area seems to me somehow redundant.
L159. Change distributional by distribution.
L161. Niche equivalence and similarity are critical concepts, but unexplained in the mss.

Methods
Workflow seems to me unnecessary.
L168. Are these 364 additional records included in Gonzales et al 2013?
L169. Please, see standards of PeerJ for citing references.
L170. Are the authors mixing data from direct sights and indirect evidence? What effect may this have on results reliability? How do the authors assign sightings or, even worst, indirect evidences to one specific lineage? This is especially worrying in the contact zone.
L177. Please, delete possible.
L178-179. Meaningless sentence.
L180. Authors may model only-climate niche, but not for dismissing other variables.
L182. used.
L182. Authors should mention climate variables originally used.
L187-188. This is confusing. Dis the authors use PC components as predictor variables in the model? If authors used original variables, how it looks, and considering that PCA builds orthogonal components, it is unnecessary to test this, and of course the figure.
L193-214. All these paragraphs are confusing, and needs deep revision and shortening.
L196. What is GHG?
L206. No information about selection criteria is provided. Authors should consider the possibility of using just two GCM, the best and the worst case ones.
L213. Authors should consider the possibility of using just one temporal scenario.
L216. Please, change are by were.
L217. I missed something, or this is the first time of considering the Chilean distribution as a whole, and the two lineages separately.
L237. Please, change use by used.
L241. Guanaco distribution models
L241. Please, consider time frame or time band, instead of time slice.
L240-241. This math is unnecessary.
L242. Averaged.
L250. Description of averaging models is inconcrete. Why do the authors use two different averaging methods? This subsection also needs clarification.
L254. Areal gains?
L257. I missed Supplemental legends. Maybe is my problem, but please revise it.
L257. Does this figure refer to surface gains as negative values?
L264. AUC <0.9 are not really good models. Please, consider that including correlated variables tends to inflate AUC values.
L277. BioSARN needs more explanation.
L285. Why do you include RAMSAR sites?
L287 and ss. I feel out of focus all the aspects related to PNC and PND. Indexes I and D are little explained. Geographic niche space is different that ecological niche space, as it depends on intensity of use.
L300. Background test needs clarification.
L302. Random points generation and what for are little explained.
L304. Why a two-tailed test? Two things may be more similar than random, but not less…
L304. Why a significance threshold of 0.1?
L305 and ss. These hypotheses are not well explained and substantiated.
L307. What are positive outcomes?
L308. Re-writing.
L308. Lack of significance does not mean hypothesis validation.

Results
L312. Original variables should be in Supplemental Material.
L315. Fig S2 and S3 are irrelevant. Besides, some correlations are >0.79, but both remain in the analysis. Why?
L317. was given.
L318. was dominated.
L318. And bio4?
L320. And bio 4, bio 7 and bio 15?
L322. Fig S4. All the variables seem to contribute similarly. How do the authors discriminate between variables? Something seems wrong when all the variables are similar in their contribution to the model.
L327 and ss. This seems somehow irrelevant to me, but it the authors do include it, it deserves an explanation.
L334. Fig 2. I can see other regions out of Chile modelled. How these locations have been selected? Why just a few ones? Guanaco was historically introduced at the Falkland Islands, why the authors do use these data, but not others in the continent?
Fig 2 shows no high quality area (just 23km2). Is not this rare, when some of the best guanaco areas in the continent are included in the modelling? This could be a mask effect of using only-climate data in the modelling.
L337. What Table?
L339-344. Comparisons between scenarios and time frames are confusing. Please, consider the possibility of using just one time frame.
L343. What are the original values? Citing Table 1?
L345-360. Results about surface changes under CC are somehow confusing. One could expect that, with an only-climate model, changes in surface after CC are going to happen. However, with this only-climate model one cannot ascertain if changes on guanaco distribution are going to happen as well, as guanaco may respond to other many factors than only climate.
L346-349. Percentages refer to threshold value, not to losses or gains, and therefore this is confusing.
L348. How do you consider significance, if it is just 23km2?
L353. I hardly see this.
L364. Please, delete Interestingly.
L372. See above comments on background test and two-tailed test.
L375. Please, see standards of PeerJ for citing references.
L379. Please, delete In spite if this.
L380. What are “important statistical differences”? This does not emerge from MaxEnt result.
L384. shows.
L387. Please delete “and”

Discussion
In general, it is rather long and little concrete.
L395. Delete comma after Although
L396 and ss. Authors should try to avoid referencing figures and tables in Discussion, except when it is essential.
L404. I feel that none of this is really extracted from your results.
L407. Please, delete For instance
L408 and ss. Is it ENSO? Is it so rare?
L413. Substantial reductions. I feel this does not emerge from your results.
L414 and ss. I do not understand how the authors relate these study cases with theirs.
L419. Maybe the authors prefer to start the sentence with In the case of…
L422-423. Pretty local references.
L424-426. Rewrite, please.
L427. See my main comments about phylogenetic-ecological relationships.
L426-427. I feel that this comes from a different methodological and analytical approach, very far from this mss.
L429. Idem.
L434 and ss. Confusing, needs rewriting.
L435. A non significant result does not allow to reject hypothesis.
L440. Please, see standards of PeerJ for scientific names.
L445. I think a significant is missing.
L447 its climatic.
L446 and ss. Again, I think that there is some confusion between evolutionary and ecological processes.
L454. Since?
L463. Maybe, the authors would consider to begin this sentence with Projected distribution models…
L467-470. This last sentence is irrelevant, and I do not share it.

Conclusions
I feel that this whole section is irrelevant for the mss, and may well be supressed.

Table 3 / Table 4
Results in the table…
Here authors mentioned a one-tailed test. Is it one or two-tailed?
I admit that Tables 3 and 4 seem to be crucial for the mss, but they are really difficult to understand.

Figure 1
Four software?
[…] in the generation of the models – What models?
The averaging procedure is little explained in the main text. Are there any references to support this? How the averaging is made? Weighted?

Figure 4
Numbers and letters in the legend do not correspond with those in the figure.

---

## Round 0.2 · Minor Revisions

Please address the point raised by the reviewer. I am not clear on the distinction noted but I favor publication with additional commentary as suggested by the reviewer.

Reviewer 1 ·

Basic reporting

All my comments in this section have been satisfactorily addressed by the authors.

Experimental design

Most of my comments have been addressed. However, my main concern regarding this manuscript still remains, namely, the use of a geographically restricted data set to model the environmental niche of the species. The authors have included a couple of sentences in the discussion regarding this issue, but I am not completely convinced it is sufficient. The authors mention that "the net effect of our conservative predictions may underestimate current and future species distribution beyond its administrative boundaries", but I don't think this is the main problem (or at least, not the one that I was trying to point out in my first review). The problem is that not capturing the entire environmental range of the species may underestimate the current and future distribution of the species WITHIN the boundaries of Chile. For instance, the species may be using a range of environmental conditions outside Chile that are currently not represented by the presence localities in Chile. Thus, when projecting the future distribution within Chile, these areas will appear as not environmentally suited for the presence of the species. This may not be too problematic if you are assessing "the evolutionary aspect of niche conservatism hypotheses", but it may be if you are estimating the area (or percentage) of range contraction and the area (or percentage) of the range in protected areas.

Validity of the findings

Again, the validity of the findings depend on how much you trust that the presence data used in the study (restricted to Chile) represent the entire environmental niche of the species. I think this needs to be more thoroughly discussed.

Additional comments

Thank you for addressing my comments. I believe the manuscript has been greatly improved. However, I still believe that an effort should be made to include the entire distribution of the species in the analysis. Have you tried obtaining data from GBIF? I know these data may have problems, but with some careful screening, reliable data can be obtained from their database.

---

## Round 0.3 · accepted · Accept

Thank you for addressing the revisions suggested by reviewers. Your manuscript has been accepted for publication in PeerJ!

# Reviewer 1 ·

Basic reporting

All my comments have been addressed.

Experimental design

The caveats included in the discussion address the concerns that I have expressed.

Validity of the findings

The caveat included in the discussion regarding the validity of the findings addresses the point that I raised in my previous review.

Additional comments

I believe the manuscript has been greatly improved and I do not have any other major comments/suggestions.